# Biophysical Properties of Bifunctional Phage-Biosensor

**DOI:** 10.3390/v15020299

**Published:** 2023-01-20

**Authors:** Vilhelmiina Juusti, Janne Kulpakko, Elizabeth Cudjoe, Ville N. Pimenoff, Pekka Hänninen

**Affiliations:** 1Laboratory of Biophysics and Medicity Research Laboratories, Institute of Biomedicine, Faculty of Medicine, University of Turku, Tykistökatu 6A, 20520 Turku, Finland; 2Aqsens Health Ltd., Itäinen Pitkäkatu 4B, 20520 Turku, Finland; 3Immunology Department, Noguchi Memorial Institute for Medical Research, College of Health Sciences, University of Ghana, Legon P.O. Box LG581, Ghana; 4Biobank Borealis of Northern Finland, Faculty of Medicine, University of Oulu, Aapistie 5B, 90220 Oulu, Finland; 5Department of Clinical Science, Intervention and Technology, Karolinska Institutet, 14186 Stockholm, Sweden

**Keywords:** phage, M13, biosensor, green fluorescent protein, GFP, liquid crystal, phage-target interaction, directed evolution

## Abstract

Biosensor research is a swiftly growing field for developing rapid and precise analytical devices for biomedical, pharmaceutical, and industrial use and beyond. Herein, we propose a phage-based biosensor method to develop a sensitive and specific system for biomedical detection. Our method is based on in vitro selected phages and their interaction with the targeted analytes as well as on optical properties that change according to the concentration of the model analyte. The green fluorescent protein (GFP) was chosen as our model analyte as it has its own well-known optical properties. Brilliant green was used as a reporter component for the sensor. Its presence enables a color intensity (absorbance) change when the analyte is present in the solution. Furthermore, the reporter dye functioned as a quencher for an additional lanthanide label in our assay. It mediated the specific phage-derived interference in the signal measured with the time-resolved luminescence. Most importantly, our results confirmed that the presented bifunctional phage with its liquid crystal properties enabled the measurement of GFP in a concentration-dependent, quantitative manner with a limit of detection of 0.24 µg/mL. In the future, our novel method to develop phage-based biosensors may provide highly sensitive and specific biosensors for biomedical or otherwise-relevant targets.

## 1. Introduction

Biosensor research is currently a fast-growing discipline to develop rapid analytical devices for biomedical, pharmaceutical, and industrial use and beyond [1,2]. Biosensing is typically based on the biophysical properties of the biosensor [3], particularly the transducer elements to convert the molecular bio-recognition into a measurable signal [4,5,6,7].

Bacteriophages, such as filamentous M13 phages, have recently been extensively exploited to develop biosensors [5,6,7,8,9,10,11] due to their unique and ideal properties such as environmental robustness, multi-function, tunable binding moiety towards the desired target [7,12], and self-assembly capabilities. For instance, the rapid colorimetric method for the detection of different bacterial species [11], fluorescent nanoparticles to detect iron ions [13], phages specifically binding colorectal cancer antigens and inducing tumor regression [12], and biosensors for *Salmonella* detection [5] have recently been developed based on M13 phages.

An important aspect of the phages is that they can exhibit liquid crystalline-like behavior [14,15,16]. Liquid crystal (LC) is a state of a particular matter formed by non-covalent interactions between molecules that induce the specific orientation of molecules within specific condensed phases. The balance of these intermolecular forces and phases is sensitive and can be easily disturbed [17]. Phage-based LCs have phase transitions between isotropic, nematic, cholesteric, and smectic phases [18,19,20]. These phases can be strongly affected by the phage concentration, temperature, and other external factors [19]. These self-assembly and liquid crystalline properties of phages have been widely utilized in functional materials [21]. Hence, phages may offer an ideal solution for developing next-generation biosensors, although it has been found to be difficult to design LC-based chemical and biological sensors that undergo controlled changes while detecting the analyte [17].

In our previous work, we developed a phage-based biosensor method capable of sensitive and specific detection of C-reactive protein and prostate-cancer-related biomarkers. The biosensor system’s optical properties change according to the concentration of the analyte and the bifunctional phage’s interactions with the target analyte and the dye molecules. Color change can be measured when the analyte is present in the solution. Furthermore, the dye functions as a quencher for the lanthanide label, which mediates the signal to the time-resolved luminescence (TRL) instrument after the phage binds to its target [22].

In this study, we assess an M13-phage-based biosensor utilizing the bifunctional phage’s interaction with its target analyte, green fluorescent protein (GFP), and brilliant green dye. The GFP was chosen as our model analyte as it is a common reporter in biophysical studies and has its own well-known optical properties [23]. The changes caused by the binding event to the absorbance spectrum and refractive index are described in relation to the luminescence measurements.

## 2. Materials and Methods

Our phage-based biosensor development protocol was published earlier [22]. In this study, the same protocol was followed with some assay-specific modifications. Stage 2 of the biopanning was performed towards green fluorescent protein (GFP, 14-392, Merck Millipore, Burlington, MA, USA) and performed with chips. The brilliant green dye (229601000, Acros Organics, Geel, Belgium) or target analyte GFP (c = 50 µg/mL) was immobilized to chips made of highly branched polymers and cellulose known as lignins. The chips were kept in a phage solution (E8111, New England Biolabs, Ipswich, MA, USA) for 15 min before washing. The host bacteria for phages, ER2738 (OD600 0.4–0.6), was directly added to the tubes containing chips in order to infect bacterial cells with dye or GFP-bound phages in a volume of 1 mL. After incubation with vigorous shaking for 3 h at 37 °C, the infected bacterial culture containing free phages was transferred to 10 mL culture for 4.5–5 h, and the chips were discarded. Altogether, five rounds of biopanning with the dye and two rounds with the GFP were performed. Six washes per round were performed with saline. Six washes were selected experimentally as a sufficient amount to gain an affinity towards the target. After the second incubation, the amplified phages with presumably increased affinity and specificity towards the dye, GFP, or both were harvested from the culture. They were then used in the next biopanning round. Finally, the developed bifunctional phages were used in the assays. The phage amount (4.0 × 10^9^ pfu/mL), GFP dilutions’ buffer concentration (0.01 × PBS and 0.2% glycerol), and label dilution concentrations (3.6 µM of Europium chloride hexahydrate, 2.1 µM of NTA and 2.1 µM of TOPO from Sigma-Aldrich, St. Louis, MO, USA) remained the same throughout the study, unless otherwise stated.

In the first assay set, brilliant green (52 µM) and bifunctional phages were added to the microplate wells. Then, dilution series with GFP concentrations of 0, 1, 2, 4, 6, 8, and 10 µg/mL were prepared and added to the microplate wells in three replicates. Additionally, either brilliant green and tryptic soy broth (TSB, Sigma-Aldrich, St. Louis, MO, USA) medium without phages (4 µL) or brilliant green and monofunctional phages and GFP dilution series were added to the microplate wells as control sets. Finally, label dilution was added to all the microplate wells. The time-resolved luminescence (TRL) (excitation, 340 nm; emission, 615 nm; integration and lag time, 400 us), luminescence (excitation, 393 nm; emission, 509 nm), and absorbance (wavelength, 623 nm) were measured from the microplate wells at the 120 min time point with a Spark multimode reader (Tecan, Switzerland).

The second assay set was performed to ensure the previously reached results in further optimized assay conditions. The protocol was otherwise the same as in the first set, but a brilliant green concentration of 36 µM was used and the GFP dilution series had a wider concentration range: 0, 5, 10, 15, 10, 25, 50, and 100 µg/mL. The time-resolved luminescence, luminescence, and absorbance were measured with the same settings as before after 60 min. Additionally, the absorbance spectrum was measured from the microplate wells with GFP dilutions of 0, 10, and 100 µg/mL around the peak area from 600 nm to 650 nm with the reader after 60 min. The absorbance spectrum was measured from the control set using TSB medium instead of phages in a similar assay.

The third and fourth assay sets were performed to study the thermotropic and lyotropic properties of the system. The brilliant green concentration was the same as in the second assay set. The GFP concentrations 0 and 25 µg/mL were used. In the thermotropic assay, similar assays were kept at +4 °C, RT, +37 °C, and +50 °C. The time-resolved luminescence and absorbance were measured with the same settings as earlier in the time points of 5, 10, 15, 30, and 60 min. In the lyotropic assay, the phage amounts of 0, 1, 2, 4, 6, 8, 10, and 12 × 10^9^ pfu were used. The time-resolved luminescence, luminescence, and absorbance were measured with the same settings as earlier in the time point of 60 min.

Finally, the fifth assay set was performed to study the system’s refractive indexes, e.g., the dependence of self-organization of phages dependent on the target. The bifunctional phages (2.0 × 10^9^ pfu) and brilliant green (52 µM) were mixed with different GFP dilutions: 0, 1, 2, 4, 6, 10, and 20 µg/mL in three replicates. The same was performed with non-specific phage having no affinity towards brilliant green or GFP. The refractive indexes were measured with LLG-uniREFRACTO 5 pro (LLG Labware, Meckenheim, Germany) after 4 h of incubation.

## 3. Results

### 3.1. Time-Resolved Luminescence

The time-resolved luminescence results were obtained from both assay sets with different GFP concentrations. The results from assay set 1 are shown in Figure 1A. The assays performed without phages and with the monofunctional phage do not show statistically significant values with any GFP concentration. Only the results with the bifunctional phage stand out with a GFP concentration of 4 µg/mL and higher. To estimate the accuracy of the detection, we observed a clear correlation: the higher the analyte concentration, the greater the detection sensitivity. The ratio was 800 between the samples where the analyte is absent and with the analyte (concentration 100 µg/mL). These results from assay set 2 are shown in Figure 1B. The limit of detection (LoD) for TRL was approximately 0.24 µg/mL, calculated from the results with assay 2.

### 3.2. Absorbance

Color formation was measured with absorbance from both assay sets with varying GFP concentrations. The results from assay set 1 are shown in Figure 2A. The sensitivity of the detection was in agreement with the absorbance and TRL. However, with absorbance, it was 0.2 and significantly smaller (inverse: 0.2^−1^ = 5) between the samples where the analyte was absent compared with the analyte (concentration 100 µg/mL). These results from assay set 2 are shown in Figure 2B. The LoD for absorbance was approximately 10.2 µg/mL, calculated from the results with assay 2.

### 3.3. Absorbance Spectrum

Absorbance was measured with brilliant green and bifunctional phage or with TSB medium as a blank in three GFP concentrations. The results with a GFP concentration of 0 µg/mL are shown in Figure 3A, with 10 µg/mL in Figure 3B, and with 100 µg/mL in Figure 3C.

### 3.4. Refractive Indexes

Refractive indexes with different GFP dilutions with both bifunctional and non-specific phage were measured. The bifunctional phage changes the refractive index in the assay due to molecular organization and analyte detection. These results are shown in Figure 4. Without the analyte, the difference in refractive index between the phages is not statistically significant (*p* = 0.1835). With a GFP concentration of 1 µg/mL, the difference in refractive index is 0.0001 (*p* = 0.0474). The difference increases when the GFP concentration is increased. The difference in refractive index between the phages was sustained as significant with GFP concentrations of 4 µg/mL (0.0002, *p* = 0.0241), 6 µg/mL (0.0004, *p* = 0.0082), and 10 µg/mL (0.0006, *p* = 0.0003). The GFP concentration of 2 µg/mL is an exception without a statistically significant result. With the highest GFP concentration, the difference between the phages is 0.0003 (*p* = 0.0021).

### 3.5. Lyotropic and Thermotropic Properties

The thermotropic and lyotropic properties of the system were studied in separate assays. Changing the phage amount and its effect on the system’s lyotropic properties and the performance of the assay with TRL is shown in Figure 5A and with absorbance in Figure 5B. Generally, the measured luminescence correlated with the GFP concentration in the assays. The influence of the phage amount on the luminescence signal measured from GFP was studied in assay set 4. The average signal above background luminescence with GFP (25 µg/mL) as a function of the phage amount is presented in Figure 6. The results of changing the incubation temperatures, affecting the system’s thermotropic properties and the performance of the assay with TRL, are presented in Figure 7A, and with absorbance in Figure 7B. When the TRL was measured, statistically significant results are achieved only at +4 °C (*p* < 0.0001) and +50 °C (*p* = 0.0230) and only after 30 min. With absorbance, the same results are achieved with measurement at +4 °C (*p* = 0.0205) after 30 min and at +50 °C (*p* = *p*-value 0.0149) after 10 min onwards. However, with measurement at +50 °C, the measurement after 60 min is not statistically significant when compared to the measurements in other incubation temperatures, except the temperature +4 °C.

## 4. Discussion

Our most important finding is that the interaction between the bifunctional phage and its analyte can be used for measuring the GFP concentration quantitatively with both absorbance and TRL. In contrast, without the phage or with the monofunctional phage, similar results were not achieved. In addition, the absorbance spectrum and refractive index results provided further confirmation of the biophysical properties of the system.

The TRL measurement is very sensitive, mostly due to the environmentally sensitive lanthanide label, which intermediates even small changes in the microenvironment to the measured signal [24] and eliminates the background luminescence [25]. This enables reliable and indirect estimation of the GFP analyte concentration in the reaction. In these assays, the sensitivity and specificity increase significantly after 30 min of incubation and the signal is measured after 60 min in this study. In our previous studies, we explored the TRL measurements with 10 or 25 min [8,26] and also 120 min of incubation [22].

The color change measured with absorbance corresponds to the analyte concentration as previously described [22]. However, these new results with GFP show a larger relative color change with a shorter measurement time (60 min instead of 120 min). Presumably, this is a consequence of better assay optimization or more suitable assay conditions than in the previous study. Additionally, the measurement data show that the color formation’s reaction kinetics halt over time. After the color stabilization, it is our understanding that this process is also irreversible. This indicates LC-like behavior, as it has been previously observed that it is impossible for LCs to disperse back to an aqueous solution after formation [14].

In addition to absorbance measurements with a single wavelength, we studied the change in the system’s absorbance spectrum due to the color formation. Previously, modified phages causing changes in the absorbance spectrums have been reported. In this manner, Sawada et al. [27] characterized plasmonic interactions between gold nanoparticles and chemically engineered peptides on the phage surface. They concluded that the interactions were fundamental for their assembly. Additionally, Koo et al. [28] demonstrated that the absorbance spectrum changes when M13 phages were embedded with silver nanoparticles.

Our main observation was that in the presence of the bifunctional phage without and with low analyte concentrations, the spectrums were almost symmetrical and narrower. Without the phage, the spectrums were wider and more irregular. The same difference is noticeable in all analyte concentrations with and without the phage. It has also been reported that the phase behavior of phages refracts and changes the measured wavelength of light [29,30]. In addition, the refractive index results show differences between the bifunctional and non-specific phage in different analyte concentrations.

The results discussed above refer to change in the microenvironment and increasing molecular stiffness in the system. Phages may share some properties with polymers that have intrinsic molecular stiffness because of their length of persistence. Hence, they have served as a model system in material science [31]. They have been used to improve the matrix stiffness [32], mimic natural bone structure [33], and enhance the stiffness of bio-laminates [34]. In turn, the phage structure is also flexible and enables the formation of different hierarchical structures via supramolecular interactions [35]. For instance, phages can form LCs due to steric forces between adjoining phages [36,37]. Overall, similar changes in the absorbance spectrum as in our system have been previously detected with LCs [38].

LCs have other commonly known characteristics such as their thermotropic [14] and lyotropic [15,21] properties. The lyotropic LC behavior of filamentous phages, such as M13, is affected by the phage concentration. The increasing phage concentration causes the self-assembly of orientated structures if it is not disrupted [21]. Furthermore, the LC behavior can be controlled by surface modification of coat proteins using chemicals [39] and fluorescent dyes [40]. However, from a technical point of view, the thermotropic liquid crystalline properties are even more important than lyotropic properties as they offer tools for various applications [14].

We performed the lyotropic and thermotropic assays to support our speculation of increasing molecular stiffness and the likely phase behavior and/or orientation in the system. The system seems to be delicate for both the phage concentration and temperature. The optimal phage amount and assay conditions are different for TRL and absorbance. Hence, the lanthanide label requires a different microenvironment to detect the changes in the system than absorbance. Our lyotropic observations are consistent with earlier findings of the phage amount’s significant effect on the phase behavior [15,21]. In addition, the temperature has a clear effect on the performance of the assay as well. The higher the temperature, the better the detection of GFP. This may be due to the improved thermodynamics in the assay and faster phase transition.

Furthermore, the bifunctional phage changes the refractive index in the system when compared to a non-specific phage. The higher the GFP concentration, the more pronounced the change; this is likely a consequence of organizing phages in the assay. Additionally, we observed that when the phage is added to the systems, it increases the luminescence signal measured from GFP. It is possible that the phages align with the molecular environment more favorably for GFP’s optical properties.

However, this study has its limitations. Before exploiting the phage-based biosensors for further use in clinical studies, the binding mechanisms of different target biomolecules should be assessed. These interactions can be studied with routine methods such as isothermal titration calorimetry and surface plasmon resonance. In addition, LC formation and phase behavior in our phage-based biosensor detection can be confirmed with optical methods not covered herein.

Altogether, the measured optical data and earlier studies support our speculation of the phages’ phase behavior in our system. In summary, phages with outstanding properties offer great potential in the development of novel biochemical and bio-optical sensors for the detection of various targets. Phage-based detection systems can be rapid, accurate, low-cost, and robust. We attempted to utilize and pursue these same characteristics while developing our system.

## 5. Conclusions

In this study, we present an M13-phage-based bifunctional biosensor for sensitive and specific detection of target molecules through optical detection. Our optical measurement data indicate that the developed bifunctional phage specifically interacts with its target analyte and the dye in the assay. That is, the presented bifunctional phage with its liquid crystal properties enabled the measurement of GFP, our model analyte, in a concentration-dependent, quantitative manner. Overall, the results herein demonstrate that our biosensor development and assay principles could be used for developing a wide range of phage-based biosensors for biomedically important targets.

## Figures and Tables

**Figure 1 viruses-15-00299-f001:**
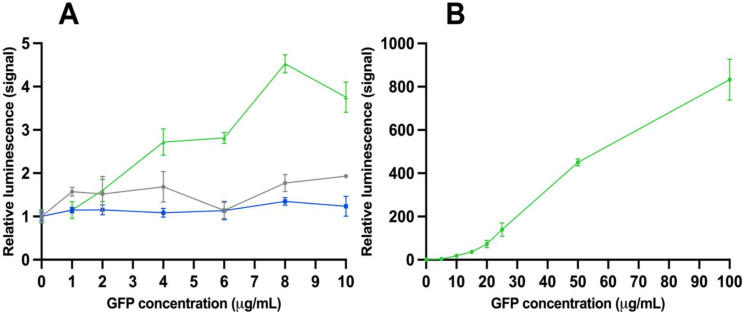
Time-resolved luminescence results from assays with bifunctional phage. (**A**) Relative time-resolved luminescence signal as a function of green fluorescent protein (GFP) concentration from the assay set 1 after 120 min. Grey line shows results from assay set 1 without phage, blue line with monofunctional phage, and green line with bifunctional phage. (**B**) Relative time-resolved luminescence signal as a function of GFP concentration from assay set 2 after 60 min. Error bars are the coefficient of variation (cv%) of three replicates.

**Figure 2 viruses-15-00299-f002:**
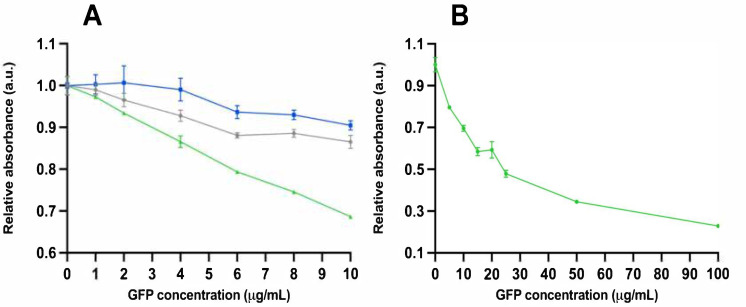
Absorbance results from assays with bifunctional phage. (**A**) Relative absorbance as a function of GFP concentration from assay set 1 after 120 min. Grey line shows results from assay set 1 without phage, blue line with monofunctional phage, and green line with bifunctional phage. (**B**) Relative absorbance as a function of GFP concentration from assay set 2 after 60 min. Error bars are the cv% of three replicates.

**Figure 3 viruses-15-00299-f003:**
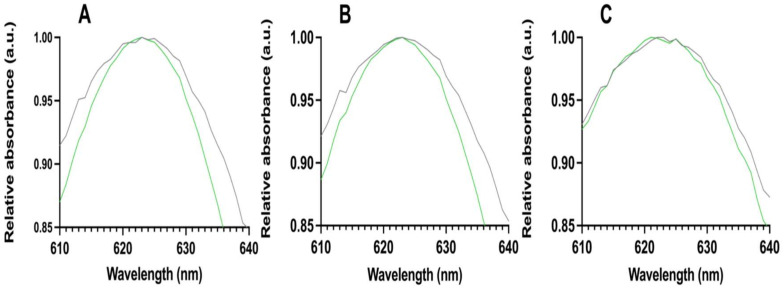
Absorbance spectrum measurements without (gray line) and with the bifunctional phage (green line). Measurements with GFP concentration of (**A**) 0 µg/mL, (**B**) 10 µg/mL, and (**C**) 100 µg/mL.

**Figure 4 viruses-15-00299-f004:**
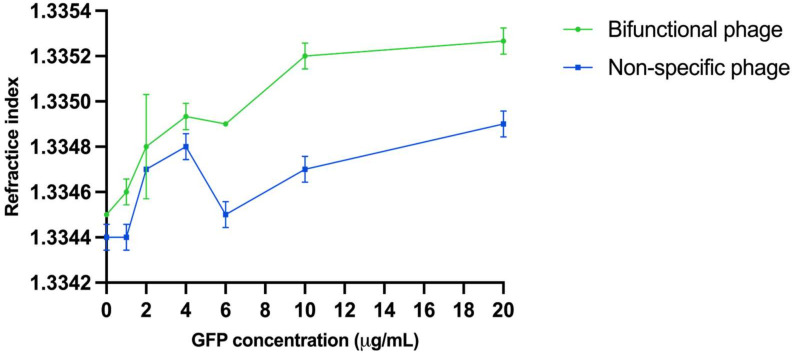
Refractive index measured as a function of GFP concentration. Blue line measured with non-specific phage and green line with bifunctional phage specific towards GFP and brilliant green dye. Error bars are the cv% of three replicates.

**Figure 5 viruses-15-00299-f005:**
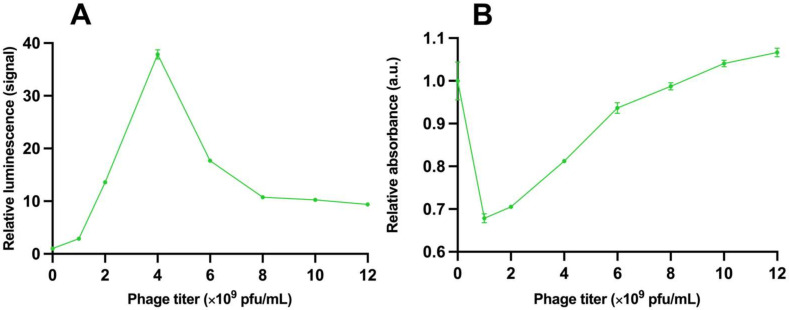
The relative ratio between GFP concentrations of 25 µg/mL and 0 µg/mL as a function of phage amount. (**A**) The ratio is measured with time-resolved luminescence and (**B**) with absorbance. Error bars are the cv% of three replicates between the concentrations.

**Figure 6 viruses-15-00299-f006:**
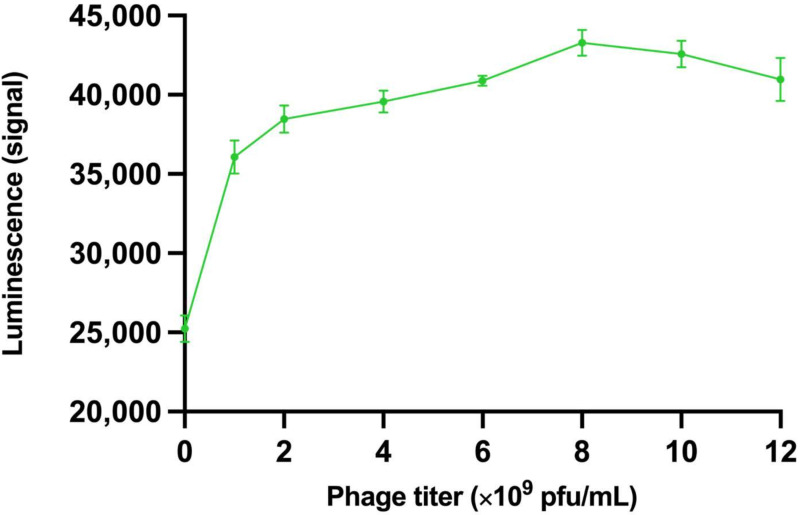
Luminescence signal above background as a function of phage amount in the assay. The GFP concentration was kept constant at 25 µg/mL. Error bars are the cv% of three replicates.

**Figure 7 viruses-15-00299-f007:**
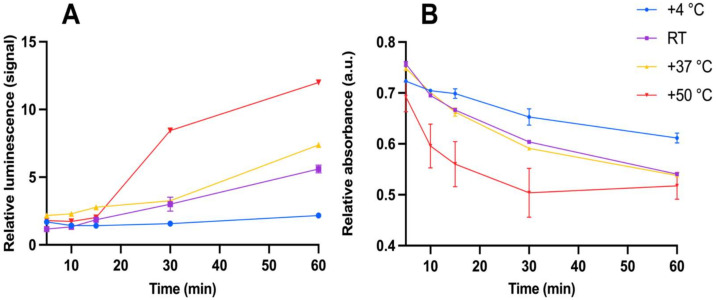
The ratio between GFP concentrations 25 µg/mL and 0 µg/mL as a function of time with incubation at different temperatures. Blue line assay incubated at +4 °C, purple line at RT, yellow line at +37 °C, and red line at +50 °C. The assay was measured with time-resolved luminescence (**A**), and with absorbance (**B**). Error bars are the cv% of three replicates between the concentrations.

## Data Availability

The data that support the findings of this study are available on request from the corresponding author (V.J.). The data are not publicly available due to commercial aims.

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
