# Peer review of "Biophysical Properties of Bifunctional Phage-Biosensor"

_viruses, 2023, doi:10.3390/v15020299_

Round 1

Reviewer 1 Report

Major Comments

1.       Abstract – results are not clearly indicated in your abstract.

Line 23 “our novel method to develop phage-based biosensors may provide highly sensitive and specific biosensors for biomedical” how can you reach into this conclusion? Have you assessed the sensitivity of your products?

2.       Materials and Methods – It is not strong. It lacks some important information.  The authors just displayed about the phage-based biosensor development. How did you propagate M13 bacteriophages?

3.       Result section – need minor improvement based on the minor comments mentioned below.

4.       In your conclusion part you have stated about the specific interaction of the developed bifunctional phage with its target analyte and the dye in the assay. How did you measure the specificity?

5.       Fig 1A and 7 statistically significant values are missed and not clearly presented in the text. The authors should show significant values in their graphs or should mention in the text.

6.       There are some grammatic errors as well as topographic errors throughout the paper (Especially spacing between words and phrases).

7.       It would be great if the author performs further additional experiment to fill the gap of some of the limitation that they have mentioned (if possible).

Minor comments

1.       Pfu should be corrected as pfu/mL in your manuscript

2.       Line 70 - (c=50 ug/mL) – unit not correctly written

3.       Line 70    “branched polymers”. What are these polymers. Not clear?

4.       Line 80 - which wells. Not clear for readers?

5.       Line 73. “Six washes per round were performed” with what? PBS or another buffer? Make it clear. Why six times?

6.       Relative absorbance measurement unit should be included in your graph (Y axis) and apply this comment for all graphs.

7.       Line 82 4 uL   - unit not correctly written

8.       Line 147 -148 ……..” We assumed that bifunctional phage changes the refractive index in the assay due to molecular organization and analyte detection. Indeed, the results in figure 4 support our assumption”. This statement seems discussion. Try to present your result accurately in numerical values supported by statistical values.

9.       Figures labeled as A, B, C etc should be properly cited in the manuscript along with the figure number for proper understanding.

10.   Line 214 – “These results” – new paragraph should not start like that

11.   Line 215 - Phages can be considered as polymers – this statement is not good. Phages may share some properties of …………..

12.   Fig 7. X axis phage amount – amount is not appropriate term – replace it with titer ….

Author Response

Response to the Reviewer 1 

Major Comments

  1. Abstract – results are not clearly indicated in your abstract.

Line 23 “our novel method to develop phage-based biosensors may provide highly sensitive and specific biosensors for biomedical” how can you reach into this conclusion? Have you assessed the sensitivity of your products?

Author’s response: The specificity and sensitivity of the biosensor is provided mainly by the specific peptide part of the phage. In the future, this can be in vitro selected further to improve performance parameters of the biosensor. For instance, we can add blocking agents in the biopanning procedure and increase washing stringency in order to improve the biosensor. 

In our previous publication, we reached sensitivity of 80% and specificity of 75% towards prostate cancer Grade Group’s (GG) 4 and 5 from urine (Kulpakko et al. 2022). The reviewed manuscript applies the biosensor development process towards Green Fluorescent Protein which acts as a model analyte for the detection. The scope was to study the biophysical properties of the system. Based on the reviewer’s comment we calculated the limit of detection for the GFP biosensor to provide more accurate results of the sensitivity. 

We revised the abstract based on your valuable comment and clarified the results of the study: “Most importantly our results confirmed that the presented bifunctional phage with its liquid crystal properties enabled the measurement of GFP in a concentration dependent quantitative manner with the limit of detection 0.24 µg/mL. In future, our novel method to develop phage-based biosensors may provide highly sensitive and specific biosensors for biomedical or otherwise relevant targets.”

  1. Materials and Methods – It is not strong. It lacks some important information. The authors just displayed about the phage-based biosensor development. How did you propagate M13 bacteriophages? 

Author’s response: We found the note important to add the needed information to the materials and methods section. The phages were propagated according to M13 Protocol for Ph.D. Phage Display (New England Biolabs) with the following modification: Lignin chips immobilized with dye or GFP (c=50 µg/mL) were kept in phage solution for 15 minutes before washing. The host bacteria for phages, ER2738 (OD600 0.4-0.6) was directly added to the tubes containing chips in order to infect bacterial cells with GFP, or dye bound phages in the volume of 1 mL. After incubation with vigorous shaking for 3 hours at 37 °C, the infected bacterial culture containing free phages was transferred to 10 mL culture for 4.5–5 hours and chips were discarded. –This step allowed us to remove and amplify chip attached phages to the growth medium and thereafter properly propagate them according to (New England Biolabs) protocol.

We have revised the propagation protocol to the manuscript as follows: The biopanning stage 2 was performed towards Green Fluorescent Protein (GFP, 14-392, Merck Millipore, USA) and performed with chips. The brilliant green dye or target analyte GFP (c=50 µug/mL) was immobilized to chips made of highly branched polymers and cellulose known as lignins. The chips were kept in a phage solution for 15 minutes before washing. The host bacteria for phages, ER2738 (OD600 0.4-0.6) was directly added to the tubes containing chips in order to infect bacterial cells with dye, or GFP bound phages in the volume of 1 mL. After incubation with vigorous shaking for 3 hours at 37 °C, the infected bacterial culture containing free phages was transferred to 10 mL culture for 4.5–5 hours and chips were discarded. Altogether, five rounds of biopanning with the dye and two rounds with the GFP were performed. Six washes per round were performed with saline. Six washes were selected experimentally as a sufficient amount to gain affinity towards the target. After the second incubation, the amplified phages with presumably increased affinity and specificity towards the dye, GFP or both were harvested from the culture. Then they were then used in the next biopanning round. Finally, the developed bifunctional phages were used in the assays. The phage amount (4.0 x 109 pfu/mL), GFP dilutions’ buffer concentration (0.01X PBS and 0.2% glycerol) and label dilution concentrations (3.6 µM of Europium chloride hexahydrate, 2.1 µM of NTA and 2.1 µM of TOPO) remained the same throughout the study, unless otherwise stated. 

  1. Result section – need minor improvement based on the minor comments mentioned below. 

Author’s response: Thank you for paying attention to the details. The manuscript has been revised based on the minor comments below.

  1. In your conclusion part you have stated about the specific interaction of the developed bifunctional phage with its target analyte and the dye in the assay. How did you measure the specificity? 

Author’s response: The specificity was studied by comparing the performance of the assays to monofunctional or non-specific phage. Similar results and detection of GFP was not observed with them. These results are shown in the manuscript figures 1A, 2A and 4. 

  1. Fig 1A and 7 statistically significant values are missed and not clearly presented in the text. The authors should show significant values in their graphs or should mention in the text.  

Author’s response: 

Fig 1A: Clarification added to the text: The results from the assay set 1 are shown in figure 1A. The assays performed without phages and with monofunctional phage do not show statistically significant values with any GFP concentration. Only the results with bifunctional phage stand out with GFP concentration 4 µg/mL and higher. 

Fig 7: Clarification added to the text: When measured with TRL, statistically significant results are achieved only in +4 °C (p < 0.0001) and +50 °C (p = 0.0230) degrees and after 30 minutes. Respectively with absorbance, the same results are achieved with measurement in +4 °C (p = 0.0205) after 30 minutes and in +50 °C (p = 0.0149) after 10 minutes onwards. However, with measurement in +50 °C the measurement after 60 minutes does not have statistical significance when compared to the measurements in other incubation temperatures except the temperature  +4 °C. 

  1. There are some grammatic errors as well as topographic errors throughout the paper (Especially spacing between words and phrases). 

Author’s response: These typos and grammatical errors have been corrected accordingly.

  1. It would be great if the author performs further additional experiment to fill the gap of some of the limitation that they have mentioned (if possible). 

Author’s response: We appreciate your comments and suggestions. Performing the additional experiments related to the limitations will remain in the scope of our future work. 

Minor comments

Author's response: Revisions made to the manuscript based on reviewer’s minor comments 1, 2, 7, 9–12: 

  1. Pfu corrected as pfu/mL 
  2. Line 70 - (c=50 ug/mL) – unit corrected 
  3. Line 82 4 uL  – unit corrected 
  4. Figures labeled as A, B, C etc revised and properly cited in the manuscript along with the figure number for proper understanding
  5. Line 214 – “These results” replaced with “The results discussed above --”
  6. Line 215 - “Phages can be considered as polymers” – replaced with the reviewer’s appropriate suggestion: “Phages may share some properties of polymers --” 
  7. Fig 7. “X axis phage amount” – amount replaced  with titer as suggested and unit corrected to pfu/mL. Same revised to the figure 5 wherein the term phage amount was also used. 
  8. Line 70    “branched polymers”. What are these polymers. Not clear? 

Author’s response: The branched polymers are the same as used in the previous study (Kulpakko et al. 2022). They are chips made of a mixture of organic branched phenolic polymers and cellulose known as lignins. The dye molecules are immobilized into them and then binding phages were selected from this surface for amplification and harvesting. This detail has been clarified in the manuscript text: “chips made of highly branched polymers and cellulose known as lignins.”

  1. Line 80 - which wells. Not clear for readers? 

Author’s response: Used wells are standard microplate wells. Clarified into the manuscript. 

  1. Line 73. “Six washes per round were performed” with what? PBS or another buffer? Make it clear. Why six times? 

Author’s response: Thank you for pointing out missing and necessary details. Washing was performed with saline that has been used as a washing solution during the biosensor development. We have experimentally observed that thus far we have not needed to increase ionic strength or detergent concentration in order to have proper biopanning results. Instead, we increased the number of saline washings. Respectively, during biosensor development we experimentally concluded to use six washes to gain suitable specificity. If the amount of washes is too small, specificity remains low and in turn, too many washing rounds will detach the target specific phages and the phage amplification will not work.  However, we are aware that different buffers can be used. It is in the scope of our future work to study the usage of buffers, detergents and different ionic concentrations in the system instead of saline. We are interested in examining how it would affect the results and biosensor sensitivity. 

We clarified to the manuscript: “Altogether, two rounds of biopanning with six washes per round were performed with saline. Six washes were selected experimentally as a sufficient amount to gain affinity towards the target.”

  1. Relative absorbance measurement unit should be included in your graph (Y axis) and apply this comment for all graphs. 

Author’s response: Absorbance unit [a.u.] added to all graphs showing absorbance data. 

  1. Line 147 -148 ……..”We assumed that bifunctional phage changes the refractive index in the assay due to molecular organization and analyte detection. Indeed, the results in figure 4 support our assumption”. This statement seems discussion. Try to present your result accurately in numerical values supported by statistical values. 

Author’s response: The sentence has been edited to be more suitable for the results section and needed numerical and statistical values provided to the text. The clarification below has been added to the manuscript: “Without the analyte the difference in refractive index between the phages is not statistically significant (p = 0.1835). With the GFP concentration 1 µg/mL the difference in refractive index is 0.0001 (p = 0.0474). The difference respectively increases when the GFP concentration is increased. The difference in refractive index between the phages was sustained significant with the GFP concentrations 4 µg/mL (0.0002, p = 0.0241), 6 µg/mL (0.0004, p = 0.0082), 10 µg/mL (0.0006, p = 0.0003). The GFP concentration 2 µg/mL is an exception with not a statistically significant result. With the highest GFP concentration the difference between the phages is 0.0003 (p = 0.0021).”

Reviewer 2 Report

The paper is very well written and proposes a new method for using phages (viruses that infect bacteria) as a basis for creating biosensors, which offer a powerful toolkit for researchers and clinicians to create sensitive and specific biosensors for a variety of biomedical and other applications. The manuscript is of potential interest to the broad readers of Viruses.

However, this study is not adequate and there are some problems, which must be solved before it is considered for publication. If the following problems are well-addressed, this reviewer believes that the essential contribution of this paper is important for bifunctional phage-biosensor.

1. Increasing sensitivity: One way to improve the sensitivity of the biosensor might be to optimize the binding affinity of the phages for the analyte, or to increase the number of phages used in the assay. Other strategies for increasing sensitivity might include the use of more sensitive detection technologies or the development of more efficient signal amplification methods.

2. Improving specificity: It may be possible to improve the specificity of the biosensor by carefully selecting the phages used in the assay and optimizing their binding affinity for the target analyte. Additionally, the use of additional reporter components or signal amplification methods may help to improve the specificity of the assay.

3. Enhancing stability: The stability of the biosensor could be improved by optimizing the storage and handling conditions for the phages and other assay components, and by developing methods to prevent degradation or denaturation of the phages or other assay components.

4. Reducing cost: One way to reduce the cost of the biosensor might be to optimize the production and purification processes for the phages and other assay components, or to find alternative sources for these materials. Additionally, the use of cheaper or more widely available detection technologies could potentially reduce the overall cost of the biosensor.

5. Expanding compatibility: The compatibility of the biosensor could be improved by developing methods for adapting the assay to different types of samples or analytes, or by developing methods for removing interfering substances from the samples.

Author Response

Response to the Reviewer 2 

Thank you for your well noted problems related to our system. Please consider our response to the problems stated. 

  1. Increasing sensitivity: One way to improve the sensitivity of the biosensor might be to optimize the binding affinity of the phages for the analyte, or to increase the number of phages used in the assay. Other strategies for increasing sensitivity might include the use of more sensitive detection technologies or the development of more efficient signal amplification methods.

Author’s response: We agree with the reviewer that the sensitivity of the biosensor is an important factor. We clarified to the revised manuscript the limit of detection (LoD) with our biosensor. LoD measured with TRF was 0.24 µg/mL and with absorbance 10.2 µg/mL. The binding affinity of the phage could be improved during the biosensor development by using buffers, detergents or different ionic concentrations in the washing step. It is in the scope of our future work to study how the usage of them instead of saline would affect the results and biosensor sensitivity. 

Currently, we use a lanthanide-label that is detected with a TRL-reader. The technology is already highly sensitive when compared to standard fluorescent measurements. Nonetheless, some signal amplification method or for example time-gated measurement might provide more sensitivity. 

The results shown in figure 5 indicate that the system is sensitive for the number of phages. Based on these results, increasing the phage amount does not necessarily improve the sensitivity. However, the assay sensitivity is delicate for the proper number of phages. The optimal phage amount and reached sensitivity depends on the biosensor as well as the detection technology. In this study, for TRL the optimal phage amount was 4.0 x 109 pfu and for absorbance reading 1.0 x 109 pfu. 

  1. Improving specificity: It may be possible to improve the specificity of the biosensor by carefully selecting the phages used in the assay and optimizing their binding affinity for the target analyte. Additionally, the use of additional reporter components or signal amplification methods may help to improve the specificity of the assay.

Author’s response: We agree that the specificity is based on finding the phages that are the best binders towards the analyte. Optimizing and increasing the binding affinity is in the best interest of our future work. It might be improved by using different washing solutions in the biopanning procedure as discussed above or by negative selection where the phages with affinity towards other molecules than target are minimized during the biopanning. Although the additional biopanning cycles could improve the specificity towards the target analyte GFP, the phages start to lose affinity towards the dye which is an important component in our system. Therefore, the number of biopanning cycles cannot be repeated too many times. In addition, currently we use a 12-mer peptide in our phage. In future, we aim to test a 7-mer peptide to improve our biosensor performance. Overall, we try to keep our biosensor very compact, and therefore scalable and cost-efficient with current reagent compositions.

  1. Enhancing stability: The stability of the biosensor could be improved by optimizing the storage and handling conditions for the phages and other assay components, and by developing methods to prevent degradation or denaturation of the phages or other assay components.

Author’s response: M13 phage as a whole (not a separate peptide produced with M13) has high stability and it can withstand different conditions. It is a clear advantage of it when compared to usage of antibodies which have a tendency to denature outside the optimal conditions. In any case, the storage and handling conditions should be studied more in order to reliably apply biosensors towards different targets and practical applications. So far, we have followed the instructions and protocols for the Ph.D. Phage Display  (New England Biolabs). Our results in figure 6 show that the assay incubation temperature has an effect on its performance. Based on our experimental results, the other assay components, the label and the dye, are relatively stable. It is worth mentioning that the label is light sensitive. However, in our system as the liquid crystal formation has occurred it can be measured even after 24 hours due to the stability of reaction. We have observed that the dye is a stable component both alone and in the system. Though we have ongoing research work related to the dye stability as well. Our aim is to keep the system as simple and robust as possible. 

  1. Reducing cost: One way to reduce the cost of the biosensor might be to optimize the production and purification processes for the phages and other assay components, or to find alternative sources for these materials. Additionally, the use of cheaper or more widely available detection technologies could potentially reduce the overall cost of the biosensor.

Author’s response: The reagent costs of our assay are moderate due to the easy amplification of phages in E. coli, simple production facilities and purification. In addition, the production and purification processes can be optimized and relatively small reagent volumes and low reagent concentrations are used. Based on the results presented in our manuscript, the assay is delicate for the equilibrium and interactions between the components and therefore we are not confident that alternative materials could be used. However, it is essential to study the options as a part of our future work. Indeed, the detection with TRL is not widely attainable because of the limited resources available. Although, we have found it to be outstanding technology for studying our system. However, because of the clear color formation as a response to the presence of analyte, it shows that especially optical measurement devices and technologies could be an attractive options for our phage-based system. If the detection only with absorbance based on the color formation would be accurate enough it would reduce the overall cost of the biosensor. All in all, the reading of our biosensor is not bound to the TRL-reader used in this study but other more affordable and portable solutions with the same technology, like portable TRL-readers, could be used.  

  1. Expanding compatibility: The compatibility of the biosensor could be improved by developing methods for adapting the assay to different types of samples or analytes, or by developing methods for removing interfering substances from the samples.

Author’s response: We think it is a relevant note that compatible biosensors would be beneficial in many different areas of detection and is a preferred feature for the biosensor. So far, we have demonstrated the system with model analytes, like C-reactive protein (CRP) and Green Fluorescent Protein (GFP), and tested it with medical sample matrices (saliva, urine). We found that the system is compatible with these matrices with simple centrifugation and dilution pretreatment of the samples. Also, we consider different filtration options for portable versions of our biosensor. The well-known properties of phages are that they have affinity for various molecules and therefore they are widely applicable. This improves the possibilities to expand the compatibility of our biosensor. 

Round 2

Reviewer 1 Report

No more comments, thank you.

Author Response

We want to thank you for the review during the second round.

Sincerely,

Vilhelmiina Juusti

Reviewer 2 Report

Accept in present form

Author Response

We want to thank you for the review during the second round. Minor spell checked has been revised to the manuscript as adviced. 

Sincerely,

Vilhelmiina Juusti